

# The first 250 years of the Heinrich 11 iceberg discharge: Last Interglacial HadGEM3-GC3.1 simulations for CMIP6-PMIP4

Maria Vittoria Guarino[1,2], Louise C. Sime[1], David Schroeder[3], and Jeff Ridley[4]

[1]British Antarctic Survey, Cambridge, UK
[2]Department of Engineering and Physical Sciences, University of Leeds, UK
[3]Department of Meteorology, University of Reading, UK
[4]Met Office, Exeter, UK

**Correspondence:** Maria Vittoria Guarino (m.guarino@leeds.ac.uk)

**Abstract.**

The Heinrich 11 event is simulated using the HadGEM3 model during the Last Interglacial period. We apply 0.2 Sv of meltwater forcing across the North Atlantic during a 250 years long simulation. We find that the strength of the Atlantic Meridional Overturning Circulation is reduced by 60% after 150 years of meltwater forcing, with an associated decrease of 0.2 to 0.4 PW in meridional ocean heat transport at all latitudes. The changes in ocean heat transport affect surface temperatures. The largest increase in the meridional surface temperature gradient occurs between 40-50 N. This increase is associated with a strengthening of 20% in 850 hPa winds. The stream jet intensification in the Northern Hemisphere in return alters the temperature structure of the ocean heat through an increased gyre circulation, and associated heat transport (+0.1-0.2 PW), at the mid-latitudes, and a decreased gyre ocean heat transport (-0.2 PW) at high-latitudes. The changes in meridional temperature and pressure gradients cause the Intertropical Convergence Zone (ITCZ) to move southward, leading to stronger westerlies and a more positive Southern Annual Mode (SAM) in the Southern Hemisphere. The positive SAM influences sea ice formation leading to an increase in Antarctic sea ice. Our coupled-model simulation framework shows that the classical "thermal bipolar see-saw" has previously undiscovered consequences in both Hemispheres: these include Northern Hemisphere gyre heat transport and wind changes; alongside an increase in Antarctic sea ice during the first 250 years of meltwater forcing.

## 1 Introduction

During the early Last Interglacial Period (LIG), a large volume of glacial meltwater was discharged from the melting Laurentide Ice Sheet (Heinrich 11 event) in the North Atlantic (Marino et al., 2015). The resulting freshwater forcing shaped the LIG climate via triggering Northern Hemisphere cooling and Southern Hemisphere warming: the thermal bipolar see-saw (Govin et al., 2012; Holloway et al., 2018). In particular, warming of the Southern Ocean during this time is attributed to a slowdown of the Atlantic Meridional Overturning Circulation (AMOC), which has been suggested as a mechanism to explain the 2-3°C Southern Ocean warming found in Southern Ocean and Antarctic climate records (Jouzel et al., 2007; Sime et al., 2009; Capron et al., 2014, 2017; Hoffman et al., 2017). Additionally, Holloway et al. (2018) suggested that this Southern Ocean warming might have contributed to sea ice decline during the LIG (Holloway et al., 2016, 2017). However, because of contrasting





model results (Holden et al., 2010; Stone et al., 2016; Holloway et al., 2018) and the difficulties of interpreting sea ice proxies

(de Vernal et al., 2013), questions about how freshwater forcing affects LIG Southern Ocean warming and whether dynamic or thermodymamic mechanisms were responsible for Antarctic sea ice loss during the LIG remain unanswered.

The AMOC is susceptible to climate change (Jackson and Wood, 2018b). Future AMOC strength is projected to decline as concentrations of greenhouse gases increase (Collins et al., 2013). The large uncertainty that accompanies Global Circulation Model (GCM) projections (Reintges et al., 2017), in terms of magnitude and time-scales of decline, makes the sensitivity of

the AMOC to different climate conditions a subject of great interest to the climate science community. AMOC changes also take centre stage in studies of past climate changes. For example, it has been hypothesized that AMOC variations might have contributed to abrupt climate change (Dansgaard-Oeschger events) in the Earth's past climate (*e.g.* Birchfield and Broecker, 1990; Sime et al., 2019).

Numerous studies have shown that freshwater release in the North Atlantic disrupts North Atlantic Deep Water formation

and the heat transport associated with the overturning circulation in the Atlantic (Rahmstorf, 1996; Goosse et al., 2002; Stouffer et al., 2006; Jackson and Wood, 2018b, a; He et al., 2020). The majority investigate the ocean response to the freshwater forcing, with the strength of the AMOC the focal point. Less attention has been given to the coupled system response, particularly wind changes that follow ocean surface cooling and subsequent wind-driven heat transport changes (Brayshaw et al., 2009; Woollings et al., 2012). This is somewhat surprising, considering that part of the upper-branch of the AMOC is wind-driven. The release

of freshwater forcing in the North Atlantic modifies directly both the ocean heat transport via water buoyancy changes, but also the atmospheric circulation via surface temperature gradient changes. Thus indirect impacts will arise from how the coupling between the atmosphere and the ocean is altered by the forcing.

Using a GCM of medium complexity, Ferrari and Ferreira (2011) showed that the heat transport in the North Atlantic can be very sensitive to mid- and low-latitude winds. In one of their water-hosing simulations, the shut off of convection in the North

Atlantic does not cause significant changes in the total heat transport because of an increase in wind-driven heat transport that compensates the loss of heat transport by convection. Ferrari and Ferreira (2011) also highlighted a lack of studies dealing with atmosphere-ocean feedbacks from freshwater release. Thus, to the best of our knowledge, only a few studies present an investigation of the atmosphere and ocean coupling in response to a release high-latitude melt-waters. Toggweiler and Lea (2010); Anderson et al. (2009) and Lee et al. (2011) investigate changes in the Southern Hemisphere winds triggered by

Northern Hemisphere cooling occurred during deglaciations and within ice ages. None of them however look at wind changes in the Northern Hemisphere and at how these changes might feedback into the total ocean heat transport by altering the gyre heat transport component. Additionally, all of these studies neglect sea ice, which is both crucial to polar changes and strongly sensitive to both oceanic and atmospheric changes.

In this study, we use the latest UK fully-coupled HadGEM3-GC3.1 climate model to simulate coupled system (atmosphere-

ocean-ice) responses during the Heinrich 11 iceberg discharge event ( ∼ 135,000 - 127,000 years ago). We investigate both the direct and indirect impacts of the Heinrich 11 iceberg discharge on the climate system. Section 2 describes methods used for setting up, running and analysing the model simulations. Section 3 contains the results of the study, in each sub-section we present and discuss the main results for the ocean (3.1), the atmosphere (3.2) and the coupled-system (3.3). Section 4 and





Section 5 conclude the study with the discussion of the results and a summary of the main conclusions. This work was carried
out in the context of the Coupled Model Intercomparison Project (CMIP6) and it is part of the Paleoclimate Intercomparison
Project (PMIP4) (Eyring et al., 2016; Otto-Bliesner et al., 2017).

## 2  Methods

### 2.1  Numerical simulations

The simulations presented in this study are run using the HadGEM3-GC3.1-LL (hereinafter HadGEM3) climate model.
HadGEM3 is the latest version of the Global Coupled configuration of the Met Office Unified Model (Williams et al., 2018).
The model consists of the Unified Model (UM) for the atmosphere (Walters et al., 2017), the JULES model for land surface
processes (Walters et al., 2017), the NEMO model for the ocean (Madec et al., 2015) and the CICE model for the sea ice
(Ridley et al., 2018). Here we use the HadGEM3 low resolution atmosphere and low resolution ocean (LL) configuration in
which the atmosphere and ocean models have a nominal resolution of 135km (atmosphere) and 1° (ocean). The UM employs
a regular latitude–longitude horizontal grid and 85 model vertical levels (terrain-following hybrid height coordinates). NEMO
employs an orthogonal curvilinear grid with a grid-spacing that decreases to 0.33° near the equator, and 75 vertical levels.
HadGEM3 was used to run all the DECK and historical CMIP6 simulations (Menary et al., 2018; Andrews et al., 2020). It
was shown to simulate very warm Arctic summers during the LIG which appear to match the observational record when run
without H11 meltwater (Guarino et al., 2020b).

Here we look at both the *Tier 1* LIG simulation (Guarino et al., 2020b), and present the first results from the HadGEM3
*Tier* 2 Heinrich 11 event (H11) simulation. These were set-up using the standard experimental protocol for CMIP6-PMIP4
*Tier1* and *Tier2* simulations (Otto-Bliesner et al., 2017; Kageyama et al., 2018). The LIG simulation was initialized from the
HadGEM3 CMIP6 Preindustrial (PI) simulation (Menary et al., 2018). To simulate the Last Interglaicial climate, HadGEM3
was forced using greenhouse gases concentration ($CO_2$, $N_2O$ and $CH_4$) representative of the Earth's atmosphere 127,000 years
ago (127ka) derived from Antarctic ice cores (for details see Otto-Bliesner et al. (2017)). The Earth's orbit at 127ka was
described using eccentricity, longitude of perihelion, and obliquity following Berger (1978). We used the same solar constant,
date of vernal equinox (21 March at noon) and all other boundary conditions (*e.g.* ice sheets, coastlines, vegetation, volcanic
activity) of the PI simulation (year 1850 fixed forcing), as per experimental protocol (Otto-Bliesner et al., 2017). The LIG
simulation spin-up period is 350 years, when the model is in quasi-atmospheric and upper-ocean equilibrium (Williams et al.,
2020). The production run for the LIG is 200 years, commensurate with model internal variability as identified by Guarino
et al. (2020a). Like the LIG production run, the H11 simulation was also initialized from the end of the LIG spin-up, running
for 250 years. Following the PMIP4 *Tier2* experiment protocol, the H11 simulation had a constant surface flux 0.2 Sv of
freshwater uniformly distributed across 50-70N within the Atlantic basin. All other boundary conditions and forcing for the
H11 simulation are identical to those applied to the LIG simulation.



## 2.2 Analysis


Climatological means are computed using the 200 year production run of the LIG simulation, and the last 100 years of the H11 simulation. Focussing on the last 100 years of the H11 simulation ensures that we allow time for the AMOC to respond to the meltwater forcing. All H11-LIG anomalies (annual, decadal and long-term) are computed against the LIG climatological mean.

Ocean Model Intercomparison Project (OMIP) diagnostics were used to calculate ocean heat budget terms, *i.e.* depth-integrated northward net ocean heat transport for each ocean basin. The 'total advective heat transport' includes transport from both resolved and parametrized advection and is the sum of the northward heat transport from gyres (here 'gyre heat transport') and the northward heat transport from overturning (here 'overturning heat transport'). See Griffies et al. (2016) Appendix I for all details. These heat budget terms are directly available from the HadGEM3 model output. Additionally, we

compute the barotropic streamfunction using CDFTOOLS: a diagnostic package for the analysis of NEMO model outputs (MEOM-group, 2021). The *cdfpsi* package was used to compute the barotropic streamfunction by integrating monthly means of the depth-integrated mass transport from South to North over the model global grid.

The annual SAM index for the H11 run was computed evaluating the zonal pressure difference between 40S and 65S (Gong and Wang, 1999). Pressure anomalies at each latitude were computed against LIG climatological values. The index was not

standardized (*i.e.* anomalies were not divided by the standard deviation of the control run), because the standard deviation of the H11 and LIG runs differ, and its unit of measure is thus hPa.

Statistical significance tests were performed using the python package for the 2-sided Welch's t-test. This type of test is more reliable for datasets not of the same size. Statistical significance is calculated using a 95% confidence interval. Note that, for readability, in our figures anomaly patterns that are statistically significant with pvalue < 0.5 are shown free of any hatches,

while areas that are *not* significant (pvalue ≥ 0.5) are hatched.

## 3 Results

### 3.1 Ocean changes

Surface cooling/warming of the Northern/Southern Hemisphere ocean surface is a robust response of the climate system to the disruption of the Meridional Overturning Circulation (Stocker and Johnsen, 2003). The addition of freshwater into the North

Atlantic reduces density and sinking at high-northern latitudes, modifying Atlantic deep water formation and density driven parts of the Atlantic meridional overturning circulation (Stocker et al., 1992; Clark et al., 1999; Knutti et al., 2004). We see an immediate response of the AMOC to the H11 freshwater forcing at 26N. An abrupt decrease of 3 Sv occurs within 25 years, followed by a sustained period of decline until around year 150 of the freshwater forcing (Fig. 1). After this period, the AMOC approaches a new equilibrium state with a mean value of the meridional streamfunction at 26N, over the last 100 years of

simulation, of ∼7 Sv. This represents a weakening of about 9 Sv, or 60 % compared to the LIG simulation.



There is a large impact of the H11 freshwater on the temperature structure of the upper ocean. The North Atlantic top ocean layer (top 200 m) cools down by ∼-0.8°C within the first 50-80 years and afterwards remains approximately constant (Fig. 2a). However, as the ocean below 200 m warms, the overall temperature of the top ∼1 km of the ocean remains approximately unchanged with a small net warming of ∼0.1°C by the end of the 250 years of H11 simulation (Fig. 2c).

This pattern of cooling and warming in the Northern Hemisphere (NH) ocean surface and subsurface layers, respectively, is consistent with the combined effects of the freshening of the North Atlantic Ocean and the slow-down of the meridional overturning circulation (Fig. 1). As the AMOC weakens, less heat is transported northward. This causes a surface cooling of the Northern Hemisphere (see Suppl. Fig. 1, see also section 3.3 for a detailed discussion on the ocean heat transport). At the same time, the enhanced freshwater flux in the North Atlantic is responsible for a freshening of the ocean surface layers

that disrupts deep convection and substantially thins the mixed layer in the region (not shown). In the absence of any vertical mixing, the colder fresher surface waters do not mix with the warmer subsurface waters and the ocean underneath is warmer in the H11 compared to the LIG (Suppl. Fig. 1), this also contributes to a further cooling of the ocean surface in the H11.

In the Southern Hemisphere (SH), whilst the top 200 m of the ocean warms by ∼+0.15°C (Fig. 2b), the top 1 km of the ocean warms by ∼+0.5°C (Fig. 2d) after 250 years. This is due to the weakened global meridional overturning circulation. We

estimate a warming trend of ∼0.2 °C/100 years in the upper 1km of the ocean (Fig. 3a). The warming trend is present at the end of 250 years, implying that system is still far from a new equilibrium. The SH warming is most intense at low latitudes and near the equator (Fig. 3b). At high-southern latitudes, cold ocean surface H11-LIG anomalies occurs at the edge of the Antarctic sea ice edge. We explore this behaviour in Section 3.3.2.

### 3.2 Atmospheric changes

The large-scale atmospheric circulation is related to the horizontal temperature gradient by the thermal wind relation. Changes in the meridional temperature gradient at the surface can influence the wind field above by modulating the strength of the vertical wind shear (*i.e.* the rate at which the wind changes with height) and the latitudinal position of the wind maxima (*i.e.* the jet stream location).

In the Northern Hemisphere, ocean surface cooling increases the pole-to-equator temperature difference and therefore the

strength of the meridional temperature gradient, compared to the LIG (Fig. 4a and 5a). The largest differences are found in the 40-55N region. At 50N, the temperature gradient is ∼-0.4 °C/100km for the H11 - this is double the LIG ∼-0.2 °C/100km gradient (Fig. 5a). During the first 10 years of simulation, SAT anomalies are weakly negative at mid- and high-northern latitudes and close to zero near the equator (dark blue curve in Fig. 6b). Over time, as negative SAT anomalies build-up in the Northern Hemisphere, the jet stream progressively increases its strength (curve colors transition from blues to reds in Fig. 6a

and Fig. 6b). The upper-level climatological zonal wind anomalies are ∼ 2 m/s stronger in the H11 than in the LIG. Wind shear associated with the mid-latitude jet stream extends to the surface (Fig. 6c): stronger H11 near-surface winds are a consequence of the jet intensification above (Fig. 4b and 5b).

The strength, shape and location of the jet stream are highly variable on a year-to-year basis. This is due to factors such as season differences, rates of tropical heating and high-latitude cooling, and stratospheric conditions. The regime can be



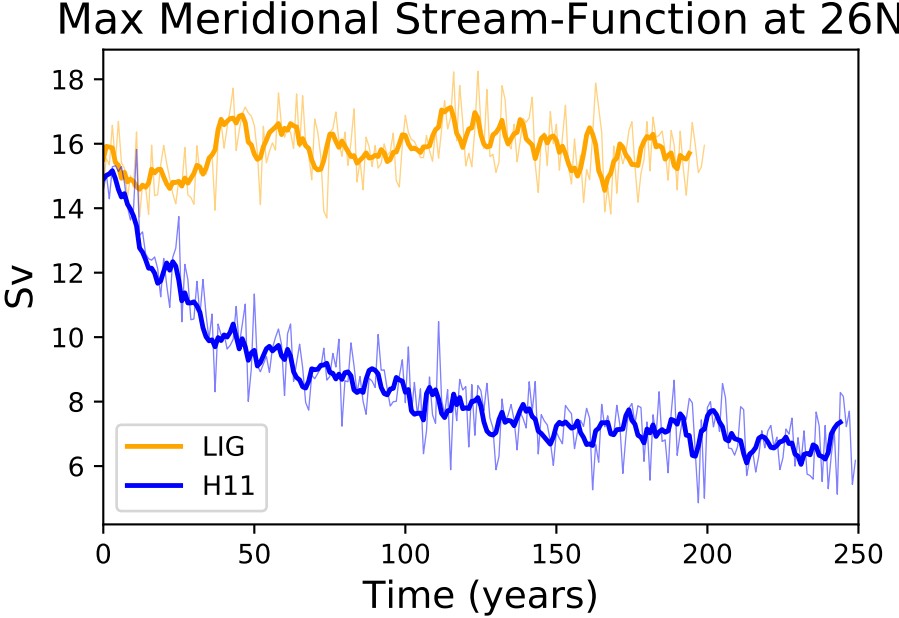

**Figure 1.** Maximum meridional stream-function at 26N for LIG (orange) and H11 (blue). Thin lines are annual means, thick lines are 11-year running means.

single- or double-jet state depending on whether the sub-tropical and mid-latitude jet are joint together or whether they are distinguishable as separate entities (Lee and Kim, 2003; Son and Lee, 2005; Lachmy and Harnik, 2014). These variations, over all time-scales, mean the climatological jet maxima is spread out approximately across 20 degrees of latitude (Fig. 6a and 6c).

     Finally, note that although we have shown here annual means only, a similar behaviour for the jet stream, the near-surface winds and the meridional temperature gradient was observed in the seasonal means. According to the jet-stream dynamics and

the fact that the pole-to-equator temperature difference is always stronger in winter and weaker in summer, the largest H11-LIG anomalies occur during the winter and spring seasons (see Suppl. Fig 2-5).

     In the Southern Hemisphere, whilst the surface warming is weak (Fig. 4a; Fig. 7b), H11 zonal wind anomalies are nevertheless significant (Fig. 7a and 7c). Zonal wind anomalies reach values of $\sim$-2-3 m/s at low- to mid-latitudes - in the 30-40S region. At higher southern latitudes, values of $\sim$+1 m/s occur.

While the surface warming observed in the Southern Hemisphere sub-tropics has the potential to generate negative wind anomalies via a mechanism similar to the one discussed above for the Northern Hemisphere, *i.e.* by decreasing the meridional temperature gradient between tropics and sub-tropics and thus weakening the sub-tropical jet (Brayshaw et al., 2008; Yang et al., 2020), the positive anomalies in Fig. 7b are too small to explain alone these simulated wind changes. We require an additional mechanism in our simulation that explains the weakened sub-tropical jet.



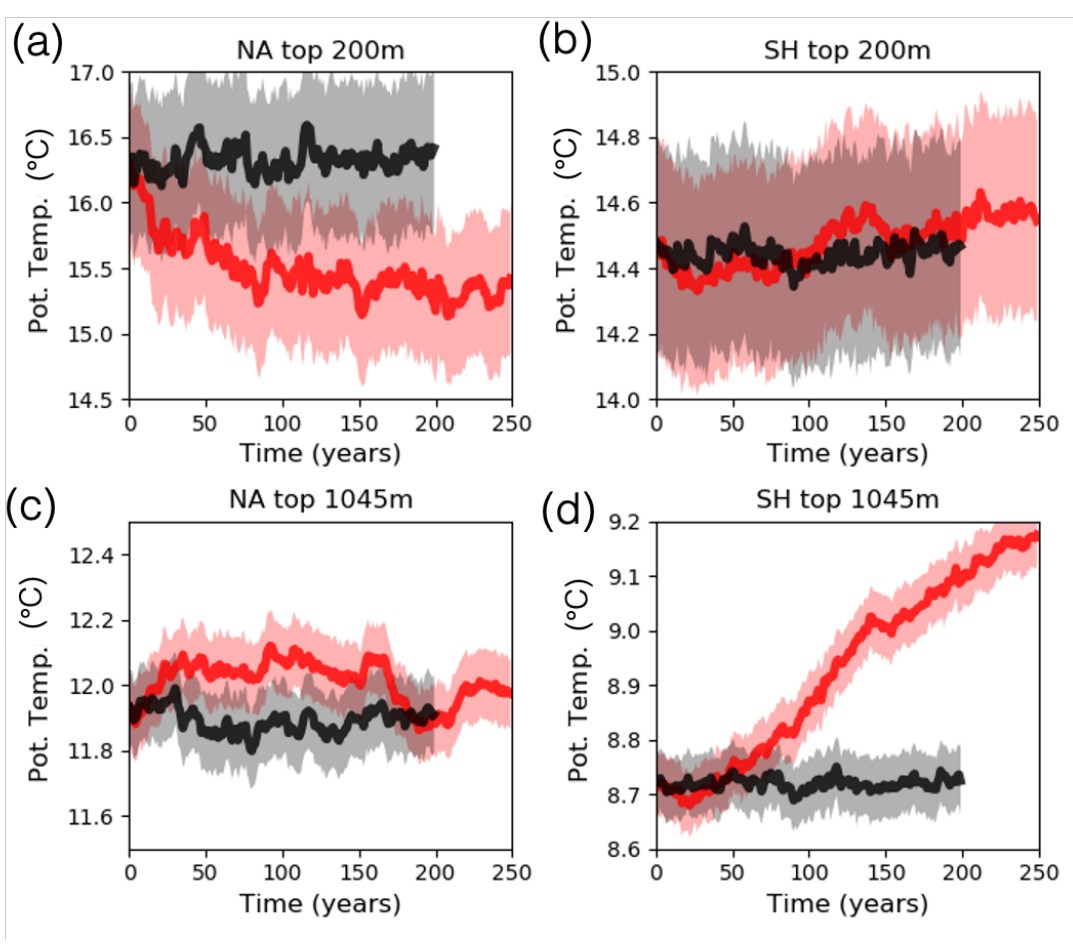

**Figure 2.** North Atlantic (NA) and Southern Hemisphere (SH) depth averaged means of sea water potential temperature for H11 (red) and LIG (black). (a) and (b) averages over the top 200m of water column, (c) and (d) averages over the top 1045m of water column. Thick lines are annual means, shaded areas represent the standard deviation.



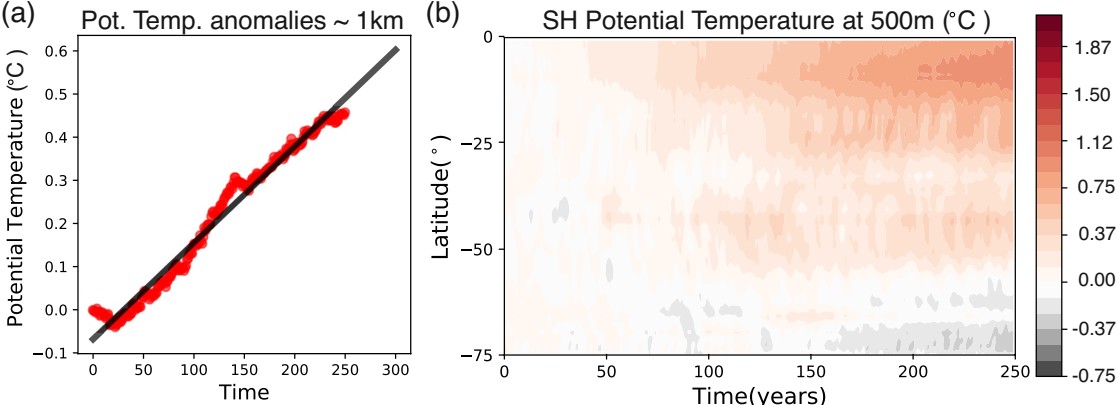

**Figure 3.** Southern Hemisphere water potential temperature anomalies: (a) depth averaged H11-LIG anomalies over the top 1045m of water column and linear fit, (b) Hovmoller diagram of H11-LIG anomalies at 500m depth.

Several studies have investigated latitudinal shifts of the Intertropical Convergence Zone (ITCZ) following a cooling at high-latitudes of the Southern/Northern Hemisphere (Kang et al., 2008; Donohoe et al., 2013; Lee et al., 2011; Ceppi et al., 2013). These have established the cause-effect relationships between temperature changes, ITCZ displacement and Hadley Cell strength. Changes in the inter-hemispheric meridional temperature (and pressure) gradient cause the ITCZ to shift towards the warmer hemisphere; in the colder hemisphere the Hadley cell strengthens because of the enhanced cross-equatorial transport of momentum, in the warmer hemisphere the other Hadley cell weakens. Here, we assess the ITCZ mean meridional position by looking at H11-LIG precipitation anomalies (Fig. 8). In the H11 simulation, the ITCZ moves southward as the Northern Hemisphere cools. The long-term mean anomalies are $\sim \pm 2$ mm/day in both Hemispheres (Fig. 8b). The southward shift of the ITCZ disrupts the SH sub-tropical jet, which is weaker in the H11 compared to the LIG (Fig. 7c).

The weakened sub-tropical jet acts to intensify the polar (eddy-driven) jet, and moves the system towards a jet-split regime (Lee and Kim, 2003). The HadGEM3 sub-polar and sub-tropical jet are distinguishable as two separate peaks in the zonal mean zonal wind (Fig.7a and 7c), with the sub-polar/sub-tropical jet stronger/weaker in the H11 than in the LIG. This strengthening of the H11 sub-polar jet has consequences for the high-latitudes surface circulation, and has impacts also on Antarctic sea ice.

### 3.3 Coupled-system responses

In the previous sections we focused our analysis on the direct impacts of the H11 iceberg discharge event on the ocean and the atmosphere. Because they originate from a perturbation applied as a boundary condition between the ocean and the atmosphere (*i.e.* the applied meltwater), the changes discussed so far can be simulated by stand-alone oceanic or atmospheric simulations. In this section, we look at how the atmosphere and ocean systems interact with each other to give rise to additional modifications of the climate system that can only be assessed in a coupled-model simulation framework.

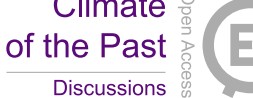

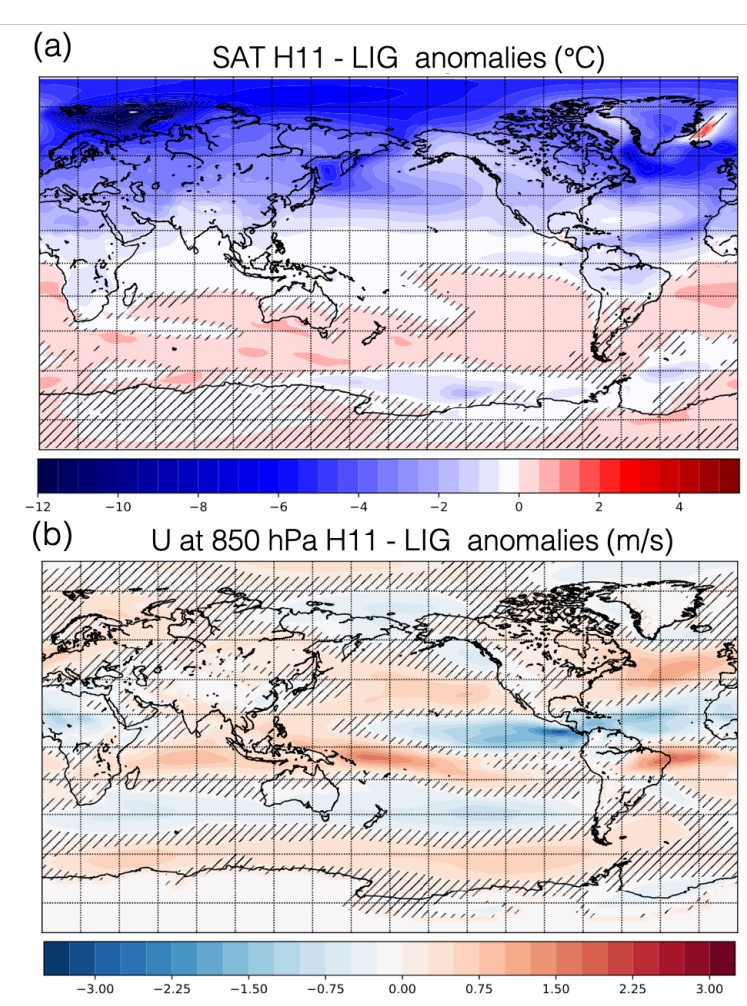

**Figure 4.** H11 – LIG Surface Air Temperature (SAT) anomalies (a) and zonal mean U at 850hPa anomalies (b). Non-hatched areas correspond to statistically significant differences (at 95% confidence)

.





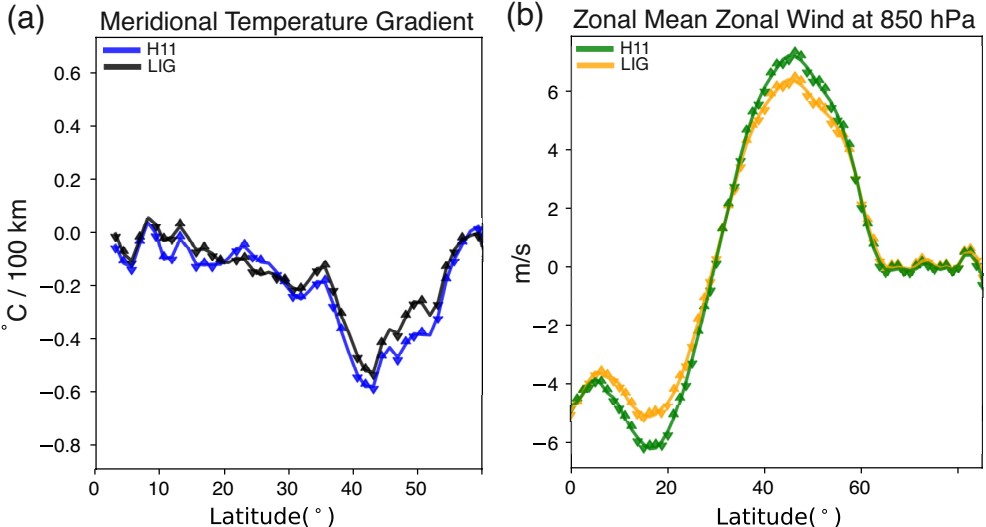

**Figure 5.** Mean Surface Air Temperature (SAT) meridional temperature gradient for H11 (blue) and LIG (black) (a) and zonal-mean zonal wind U at 850 hPa for H11 (green) and LIG (orange) (b) for the North Atlantic (80W – 10W) region. Solid lines are annual means, error bars are the standard error of the mean.

### 3.3.1 Heat transport changes

In the H11 simulation, the total advective heat transport decreases everywhere in the Atlantic basin (Fig. 9b). The major drop ( ∼-0.25 PW) occurs during the first 50 years of simulation (Fig. 9c and Suppl. Fig. 6a). The temporal evolution of the Atlantic advective heat transport resembles the AMOC trend (Fig.1), with values reaching a plateau ( ∼0.4 PW, or -33 % change) near year 150 of simulation. In the Northern Hemisphere, the largest decrease is ∼ -0.4PW at about 10N. (Fig.9b) In the Southern Hemisphere, the difference in Atlantic advective heat transport between the H11 and LIG simulation is about -0.25PW between

the equator and 35S (at the end of the Atlantic). This means that some combination of the Indian, Pacific, and Southern Oceans are taking up an addition 0.25PW of ocean heat transport in the H11 simulation that was being lost in the North Atlantic in the LIG simulation. This explains the H11 warming at the surface (Fig.4a) and at 500m depth (Fig.3b) at those latitudes, since extra heat is being stored by the ocean.

The overturning heat transport component (Fig.9e-f), which is largely density driven, is the major contributor to the overall

decrease in Atlantic advective heat transport (also referred here as total northward heat transport). This component is directly linked to the strength of the meridional overturning circulation and its decrease is uniform across all latitudes. On the other hand, the wind-driven gyre heat transport component exhibits a far less uniform trend (Fig9.h-i).

Northern Hemisphere westerly winds are about 20% stronger in the H11 simulation. The maximum positive zonal wind anomaly is centred in the 40-50N latitude band (Fig. 5b), consistent with the jet-stream intensification (Fig. 6c). This is roughly

where the boundary between the subpolar and the subtropical gyres is located (Fig. 10). The barotropic stream function weak-



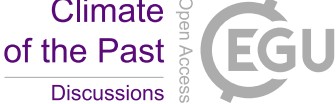

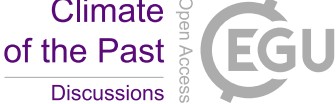

**Figure 6.** Each line in panel (a) corresponds to a 10-year mean of the zonal mean zonal wind U at 300 hPa for the H11 run. Similarly, in panel (b) each line represents a 10-year mean of H11-LIG Surface Air Temperature (SAT) anomalies. Panel (c) shows the long-term mean of zonal mean zonal wind H11-LIG anomalies. Non-hatched areas correspond to statistically significant differences (at 95% confidence).



**Figure 7.** As in 6 but for the Southern Hemisphere.

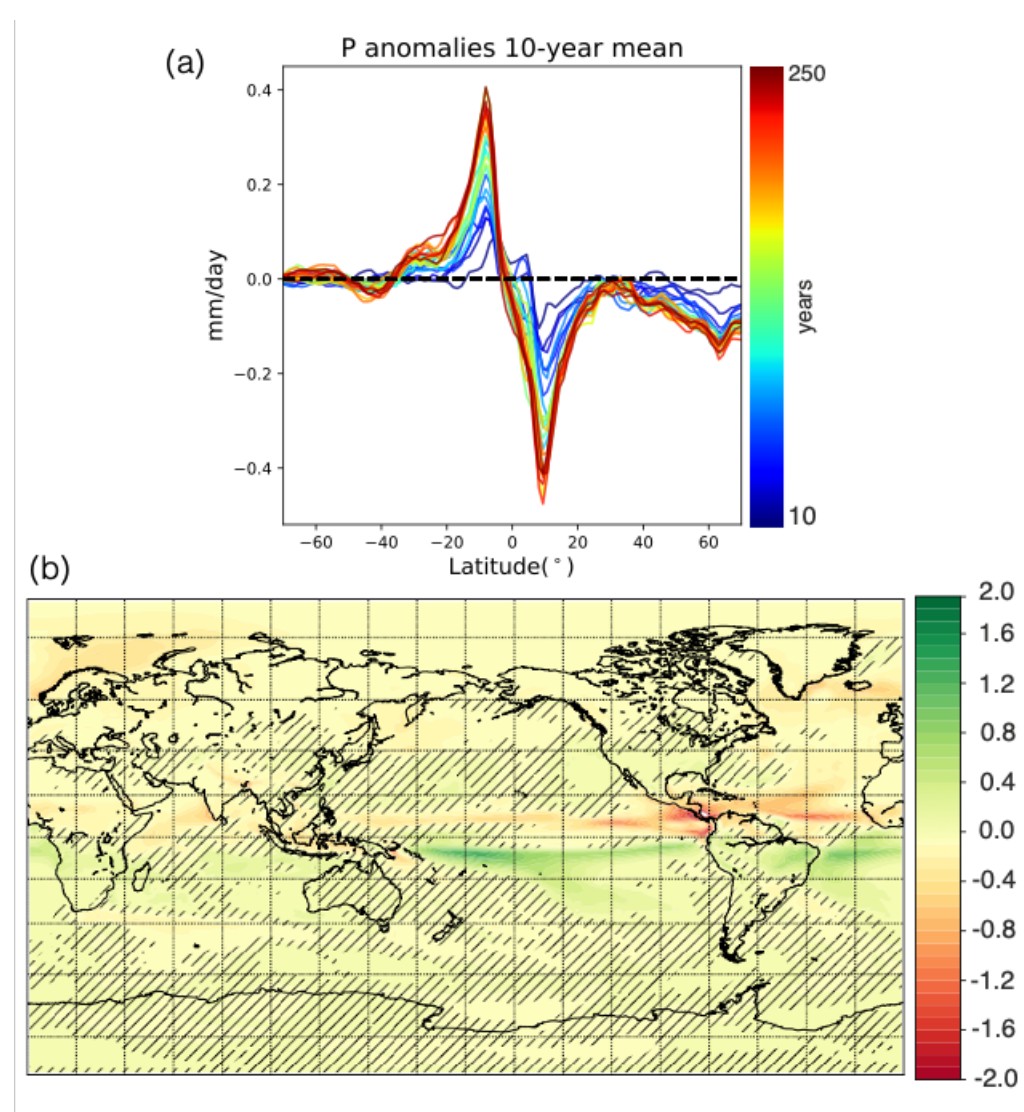

**Figure 8.** Each line in panel (a) corresponds to a 10-year mean of H11-LIG Precipitation (P) anomalies. In panel (c) the long-term mean of P anomalies is shown. Non-hatched areas correspond to statistically significant differences (at 95% confidence).





ens in the subpolar gyre for the H11 simulation because the enhanced westerlies act to decelerate the contour-clockwise rotation (*barosf* negative) of the gyre. At the same time, the subtropical gyre intensifies as the stronger westerly winds favour the clockwise (*barosf* positive) rotation of the gyre in that region. Because of the location of the maximum wind anomaly between 40-50N, the stronger westerlies impact more greatly the subpolar than the subtropical gyre in terms of gyre shape. The south-

ern branch of the subtropical gyre is also affected by wind changes. The barotropic streamfunction for the subtropical gyre strengthens in the 20-30N region (particularly in the western quadrant) because of the modest acceleration in the easterly trade winds (Fig. 5b). The net result is an overall weakening of the subpolar gyre and an intensification of the subtropical gyre in the H11 compared to the LIG (Fig. 10). This results in a decrease in the heat transport due to the gyre circulation of ∼-0.2 PW at high-northern latitudes between 50-60N, but an increase of ∼+0.1-0.2 PW over mid-latitudes in the 20-50N region (Fig. 9h).

The weakening/strengthening of the subpolar/subtropical gyres approximately balance in terms of total gyre heat transport: the northward heat transport due to the gyre circulation in the North Atlantic is nearly unchanged by the H11 forcing (Fig.9i). However the implications of a different regional distribution of ocean heat are of significance. If the response of the coupled-system, *i.e.* changes in the gyre heat component were not simulated, a larger reduction in total advective heat transport would occur at mid-latitudes. Furthermore, the decrease in total advective heat transport at high northern-latitudes (Fig.9b), north of

50N, would be non-existent. This is because the overturning heat transport term is zero above 50N (Fig. 9e), *i.e.* the decrease comes from the weakening of the sub-polar gyre (Fig. 9h). This weakened sub-polar gyre thus contributes to the large ocean surface cooling in the North Atlantic (Fig. 4a). This process is not usually identified when models have been used to explain the cooling of the Northern Hemisphere during Heinrich events.

The changes in the Atlantic basin heat transport terms discussed above dominate the signal also in the global ocean heat

transport terms (Fig.9 a,d,g). The opposite response of the sub-tropical and sub-polar gyres is still present when the global gyre heat transport term is computed. As for the overturning component, this is still the main responsible for the decrease in global heat transport, but south of 10N heat gain in the Pacific reduces LIG and H11 differences in the overturning and advective global heat transport terms. South of ∼55S there is no difference in the global advective heat transport for the two runs. This indicates that in our 250-year long simulation the freshwater forcing result in heat accumulation in the Southern Ocean between

the equator and ∼55S, but no heat transport (and no heat accumulation) southward of 55S (see also Discussion).

### 3.3.2 Southern Hemisphere sea ice changes

The net effect of a weakened Southern Hemisphere sub-tropical jet (caused by the southward shift of the ITCZ) is to intensify the Southern Hemisphere polar jet and surface winds (Fig. 11ab), which results in a more positive Southern Annular Mode (SAM) (Fig.11c). Winds are stronger in the H11 simulation than in the LIG, with wind anomalies at 850 hPa up to 1 m/s (Fig.

11b). As the belt of westerly winds become stronger, the SAM becomes more positive. The mean value for the SAM index (non-standardized) is -0.05 hPa for the first 150 years of simulation (Fig. 11a) and 0.4 hPa for the last 100 years of simulation (Fig. 11b), resulting in a positive trend for the annual SAM index (Fig. 11c).

Here we find that in response to a increasingly positive SAM, in our H11 simulation the Southern Hemisphere sea ice expands with maximum H11-LIG sea ice concentration (SIC) anomalies of ∼+20% (Fig. 12).





The connection between a positive SAM and an increase in Antarctic sea ice is widely reported in the literature (*e.g.* Hall and Visbeck, 2002; Lefebvre et al., 2004; Ferreira et al., 2015; Turner et al., 2015; Holland et al., 2017)). A positive SAM influences sea ice formation and growth both dynamically and thermodynamically. The stronger westerly winds advect sea ice away from the coastlines more efficiently and thus increase the ice concentration along the ice edge (Fig.12) (Hall and Visbeck, 2002; Lefebvre et al., 2004). At the same time, the enhanced westerlies cause an anomalous equatorward Ekman flow that,

advecting colder water from the South, decreases the sea surface temperature and promotes sea ice formation (Ferreira et al., 2015; Holland et al., 2017).

We note however that the typical response of sea ice to a positive SAM is a dipole pattern of anomalies, for which sea ice decreases in the Weddell sea sector and increases in the Amundsen and Ross Sea sectors (Lefebvre et al., 2004; Simpkins et al., 2012). This pattern is not visible in the annual mean results (Fig. 12), where rather sea ice increases along the ice edge both in

the Weddell and Amundsen and Ross Sea sectors. This is however partly due to seasonal dependence of the dipole, for which the pattern is most visible in the winter months (Simpkins et al., 2012), rather than in the annual mean.

Furthermore, upwelling of cold waters in the Southern Ocean between 40S and 70S has been postulated as explanation for the increase of sea ice in the Weddell sea sector in response to a North Atlantic cooling Crowley and Parkinson (1988). This mechanism was invoked by Renssen et al. (2010) to explain positive SH sea ice anomalies during the Early Holocene

deglaciation (∼9,000 years ago). In our simulation the ocean zonal mean temperature anomalies are different from Renssen et al. (2010): at depth, the Southern Ocean warms near 40S (Fig.3 and Suppl.Fig.1). We also found either zero or very small negative temperature anomalies between 50-70S at the surface (Suppl. Fig. 1). This implies that some very weak upwelling of cold waters might be happening in the HadGEM3 model and could also contribute to the sea ice increase in the Weddell sea sector. The very weak upwelling of cold waters in the Southern Ocean in our simulation is supported by the fact that the

expected lag between North Atlantic cooling and SH cold waters upwelling is of 100-200 years (Goosse et al., 2004; Renssen et al., 2010) but HadGEM3 was run here for 250 years.

## 4   Discussion

The observational record from the LIG suggests that by around 128 ky, after a 4-5000 year H11 event, there should be 2-3°C ocean warming at high southern latitudes (Capron et al., 2014, 2017; Hoffman et al., 2017). The positive warming trend of

∼0.2 °C/100 years of Fig.3 implies that, after 250 year of H11 forcing, the Southern Hemisphere keeps on warming in our H11 simulation (see also section 3.1). If one extrapolates, using a linear fit to Fig.3, after ∼ 1500 years the model may have simulated 2-3°C warming. In this regard, our results are somewhat similar to previous studies like Holloway et al. (2018) who ran a 1,600 years long simulation with the older HadCM3 model and obtained a ∼+1.5°C warming. However, we are cautious about extrapolating thousands of years from a 250 year simulation: this linear trend may not be sustained in the long-term

towards equilibrium.

**Figure 9.** Total Advective (a,b,c), Overturning (d,e,f), and Gyre (g,h,i) northward heat transport for the Global basin (a, d, g), the whole Atlantic basin (b, e, h), and the North Atlantic basin only (c, f, i) for the H11 and LIG simulations. Panels a-b, d-e, g-h show zonal means computed over the long-term means. Panel c,f,i show area-weighted annual timeseries.

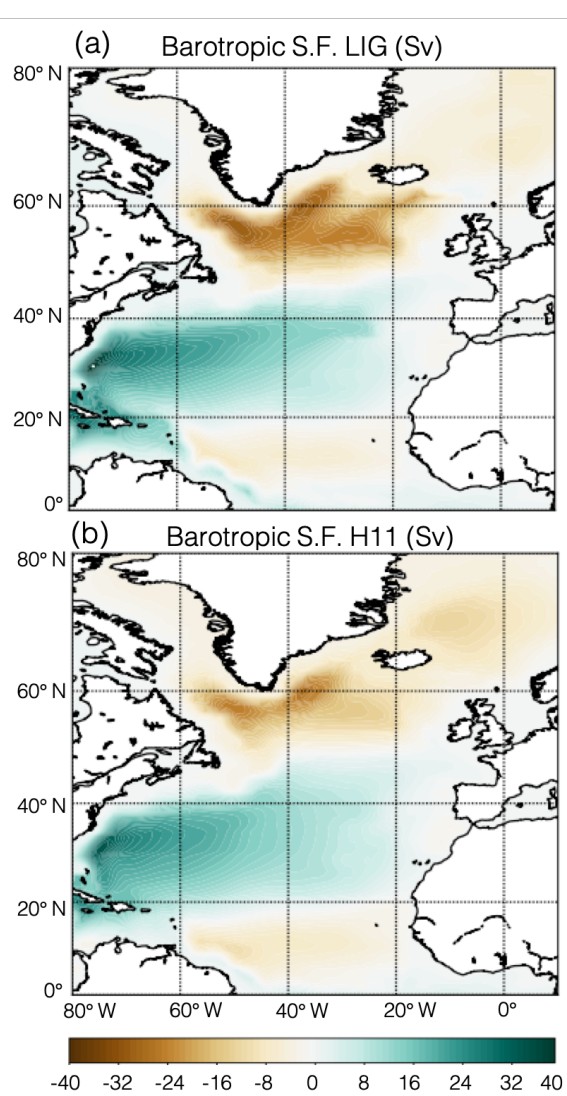

**Figure 10.** The annual mean barotropic stream function in the North Atlantic for the LIG (a) and the H11 (b) simulations.





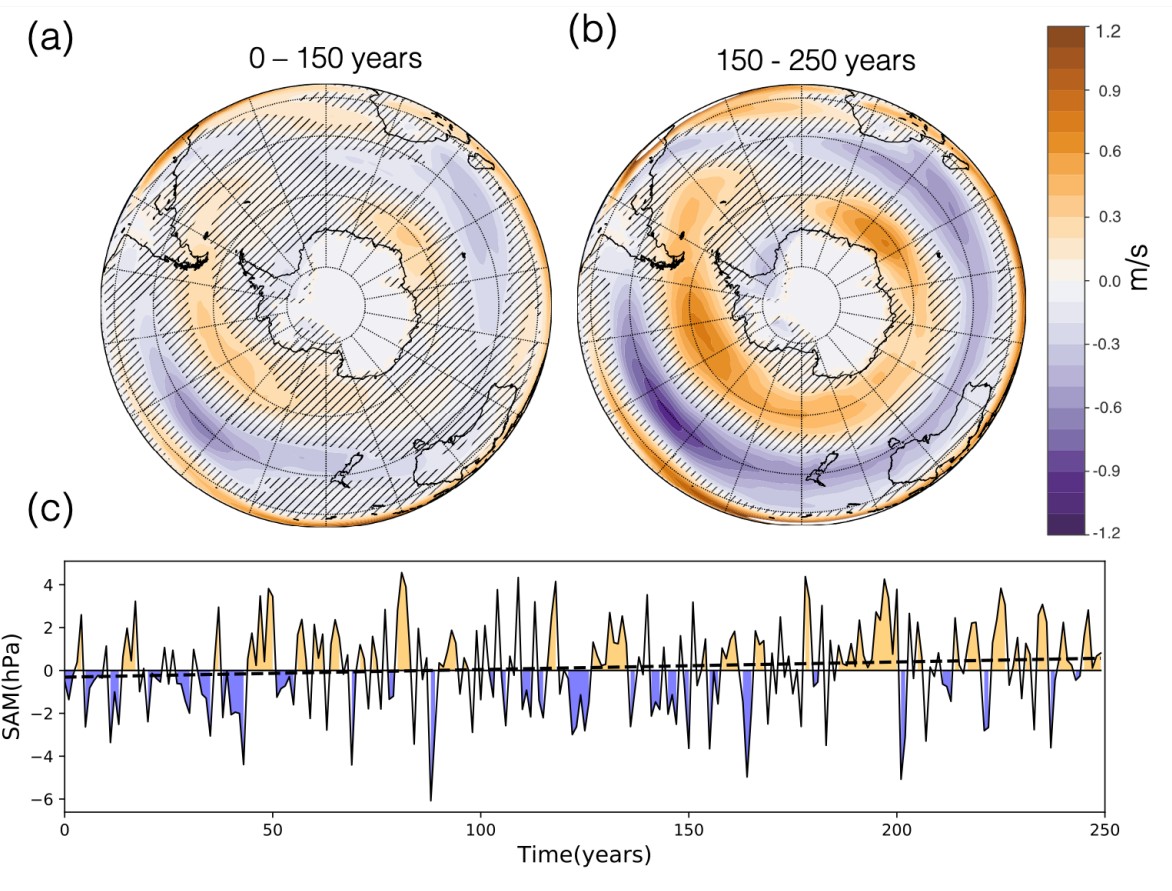

**Figure 11.** H11 – LIG anomalies for U at 850hPa computed over the first 150 years (a) and the last 100 years (b) of simulation. Non-hatched areas correspond to statistically significant differences (at 95% confidence). The annual SAM index for the H11 simulation is shown in (c). The linear regression line (dashed) represents a statistically significant positive trend with pvalue < 0.05.

In our H11 simulation, the warming of the Southern Ocean is driven by a redistribution of ocean heat. After 150 years of simulation the AMOC is reduced by around 60%, therefore considerably less heat is being lost in the NH each year, which leads to a continual gradual build of heat elsewhere in the ocean.

The changes in the North Atlantic heat transport terms (gyre and overturning components) discussed in section 3.3.1 are the
main responsible for the NH cooling and SH warming, and dominate the signal also in the global ocean heat transport terms (Fig.9). It is worth pointing out that previous studies have associated the asymmetry between the sub-polar and the sub-tropical gyre, similar to the asymmetry observed here, to a leakage of freshwater from sub-polar latitudes into sub-tropical waters (Swingedouw et al., 2013). While a leakage of freshwater signal cannot be ruled out for the HadGEM3 model (Menary et al., 2018), in previous hosing experiments showing leakage the intensity of both the sub-polar and sub-tropical gyres decreased
over time (with the sub-tropical gyre only marginally affected), and the freshwater forcing was exclusively applied along the





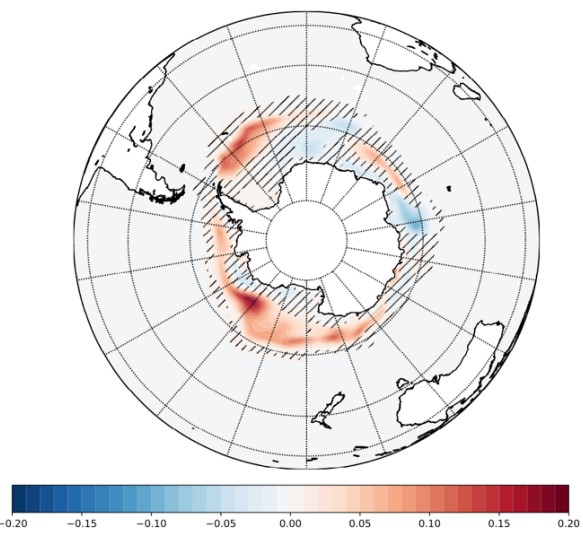

**Figure 12.** Southern Hemisphere H11 – LIG sea ice concentration anomalies. Non-hatched areas correspond to statistically significant differences (at 95% confidence).

Greenland coastline (Swingedouw et al., 2013). Our uniform release of freshwater in the 50-70N latitudinal band should not encourage freshwater leakage through the oceanic pathways described by Swingedouw et al. (2013) and, most importantly, the strengthening of the sub-tropical gyre in our H11 simulation is a major difference from Swingedouw et al. (2013) corroborating that alterations to the gyre circulation in the H11 run are driven by wind changes.

Regarding atmospheric changes, the positive SAM and the larger sea ice concentration found in our H11 simulation are findings consistent with each other, and are in agreement with the literature on the subject. However, positive SIC anomalies for our H11 simulation put our findings in contrast with the hypothesis that, during the Last Interglacial, the SH sea ice began to decline following the Heinrich 11 event (Holloway et al., 2018). This hypothesis is based on the assumption that ocean heat would build up in the Southern Ocean in response to the weakened overturning meridional circulation.

Over the first 250 years of their H11 simulation, Holloway et al. (2018) show that in HadCM3 (older UK model), SH winter sea ice goes into immediate decline, and after 200 model-years sea ice area anomaly is ∼-10 % (see their Fig. 1). This older HadCM3 model has a rather simplified representation of the atmosphere and the ocean compared to HadGEM3 (HadCM3 is the predecessor of HadGEM3 with around 20 years of model development between them).

     In our 250-year long simulation, while freshwater forcing has immediately a major impact on the (North and South) Atlantic
basin (Fig.9b), the global heat transport south of 55S remains unaffected (Fig.9a). Thus, in our simulation sea ice does not respond to changes in ocean heat transport (yet) but rather to shorter time-scale forcing such as the SAM. This is agreement with previous studies showing the existence of an approximately 200-year lag between changes in freshwater forcing from Greenland and the onset of warming in Antarctica since the warming signal takes time to cross the Antarctic Circumpolar Circulation (at about 50-60S) (Buizert et al., 2015; Pedro et al., 2018; Svensson et al., 2020).



As described in section 3.3.2, a more positive SAM in the H11 run causes an increase in SH sea ice in HadGEM3. However, on longer time-scales it is possible, and perhaps likely, that sea ice might start responding to a more pronounced warming of the Southern Ocean (according to trend of Fig.3) and that could eventually start declining: a much warmer Southern Ocean will inhibit sea ice formation and will contribute to faster sea ice melt. Additionally the SAM may shift into a different state. For example, the weak sub-tropical warming observed at the ocean surface in our H11 simulation has the potential to further

weaken the sub-tropical jet, and move the system to a proper jet-split regime in which the sub-tropical and the sub-polar jet are distinguishable. This would result in an even stronger sub-polar jet and thus a more positive SAM which might reinforce the role of the SAM in promoting sea ice increase. However, only a longer simulation would allow a proper investigation of such mechanisms.

Other aspects which would be worthy of further investigation (with a longer simulations) include:

– If the Southern Ocean continues to warm towards + 3°C warming, the pattern of warming will be of importance (Brayshaw et al., 2008). An homogeneous warming that extends to the higher-latitudes might disrupt and weaken the sub-polar jet (by decreasing the polar to mid-latitudes meridional temperature gradient). Under this scenario, the SAM will polarize towards its negative phase and sea ice could decrease;

– Even with a SAM persistently positive, a two time-scales evolution of sea ice is possible. First proposed by Ferreira et al.
(2015), following an initial sea ice increase (according to the mechanisms discussed here), on longer time-scales sea ice is expected to decrease when the SAM is positive. This is because, on long time-scales, the deeper ocean circulation is also affected by the changes in the wind forcing. In particular, the upwelling of deeper and warmer ocean waters will eventually increase the SST and melt the sea ice.

To conclude, we hypothesize that the positive SIC anomalies in Fig.12 are partially due to the limited length of our simulation
(250 years). It is well known that the ocean system responds to forcing on time-scales that can be longer than thousand years. We speculate that, for a longer H11 simulation, the weak warming signal currently observed in the Southern Hemisphere might become larger and a more significant warming of the Southern Ocean might be simulated (according to Fig. 3).

In this perspective, the positive trend in the SAM and the slowdown of the AMOC might be thought of as two competitive mechanisms that cause sea ice to increase/decrease depending on the process that is dominating the sea ice response. Our
findings thus suggest that there might be a transient system response for which sea ice actually increased its extent during the LIG for a few hundreds years (or more).

However, without prolonging our simulation it is not possible to investigate how these two mechanisms might interact with each other. Thus, whilst a longer H11 simulation using our model is difficult due to the prohibitively high computational cost, supercomputing advances are needed to enable a more in-depth investigation of some of mechanisms described here for
Southern Ocean warming and Antarctic sea ice increase. Until this is possible, longer term H11 hemispheric teleconnections, Southern Ocean warming, and when and how Antarctic sea ice disappeared during the Last Interglacial may remain somewhat elusive.



## 5    Conclusions

In this study we have analysed the response of the ocean and the atmosphere to an enhanced release of glacial freshwater within

the North Atlantic basin during the Heinrich 11 (H11) event in the Last Interglacial (LIG) period (∼135 -128 thousands years ago).

We used the UK CMIP6 model (HadGEM3) to simulate a time-slice of the Last Interglacial climate at 127,000 years ago (Guarino et al., 2020a). A constant flux of freshwater equal to 0.2 Sv was added to the North Atlantic between 50-70N, and the model run forward for 250 years to simulate the H11 iceberg and meltwater event. This simulation was carried out adhering to

the international protocol for the PMIP4-LIG Tier 2 simulations (Otto-Bliesner et al., 2017).

After 150 years of freshwater forcing, the Atlantic Meridional Overturning Circulation (AMOC) is reduced by about 60 %. After a steady decline, and for the last 100 years of simulation, the AMOC remains approximately constant at around 9 Sv.

The combined action of a fresher North Atlantic Ocean and a weaker meridional overturning circulation leads to a pattern of cooling (-0.6°C)/warming (+0.25°C) in the Northern Hemisphere ocean surface/subsurface that is in agreement with previous

studies (Stocker et al., 1992; Clark et al., 1999; Knutti et al., 2004; He et al., 2020). In the Southern Hemisphere, both the ocean surface and subsurface warm, with an estimated trend of ∼0.2 °C/100 years in the upper 1km of the ocean. This trend remains approximately linear and constant for 250 years. After 250 years (length of simulation) the warming trend is continuing and the system has not reached a new equilibrium.

The pole-to-equator temperature difference increases in the H11 compared to the LIG in the Northern Hemisphere due to

polar cooling. The largest increase in the meridional surface temperature gradient between the H11 and LIG simulations occurs between 40-50N. These changes in the meridional temperature gradient at the surface influence the wind field. In the H11, 850 hPa winds are found to be ∼20 % stronger than in the LIG with a maximum positive anomaly centred at 45N. This is a direct consequence of the intensification of the jet-stream above that is ∼ 2 m/s stronger in the H11 than in the LIG.

The jet intensification in the Northern Hemisphere alters the total northward ocean heat transport (*i.e.* the advective heat

transport) by increasing and decreasing the gyre circulation at mid- and high-latitudes respectively. At mid-latitudes, increased subtropical gyre circulation means that this part of the ocean circulation transports more heat northward. At the same time, the weakening of the overturning circulation in the same region reduces northward overturning heat transport. Whilst these two elements act in opposite directions, the overturning component effect dominates, so the total advective heat transport decreases in the H11 compared to the LIG. At higher latitudes in the Northern Hemisphere, the contribution to the total advective heat

transport from the overturning circulation is zero but in this case the weakening of the subpolar gyre makes the total advective heat transport decrease in this region for the H11 simulation.

As the Northern Hemisphere cools down, the changes in meridional temperature and pressure gradients cause the Intertropical Convergence Zone (ITCZ) to move southward with precipitation anomalies equal to ∼ -/+ 2 mm/day in the Southern/Northern Hemisphere. In the Southern Hemisphere, the shift of the ITCZ disrupts the SH sub-tropical jet. To a weakening

of the sub-tropical jet (∼ - 2-3 m/s ) corresponds a strengthening of the sub-polar jet (∼ +1 m/s) and the system moves towards a jet-split regime.

The SH sub-polar jet intensification leads to stronger westerlies and to Southern Annual Mode (SAM) changes. The SAM exhibits a positive trend and becomes overall more positive during the last 100 years of H11 simulation. A positive SAM influences sea ice formation both dynamically and thermodynamically, and acts to increase Antarctic sea ice in the H11 compared to the LIG.

Our coupled-model simulation framework shows for the first time that the classical "thermal bipolar seesaw" associated with the H11 event has undiscovered consequences in both Hemispheres: these include Northern Hemisphere gyre changes, alongside an increase in Antarctic sea ice during the first 250 years of meltwater forcing. We speculate that the SH sea ice increase that our model newly captures may be part of a two-stage Antarctic sea ice response to the H11 event (see Discussion). Highly resolved records of Last Interglacial changes from the Southern Ocean could be examined for evidence of this occurrence. This knowledge should also be of value to the paleoclimate proxy community in interpreting current contrasting sea ice observations (*e.g.* Chadwick et al., 2020).

Finally we note that the consequences of the applied meltwater forcing on the ocean gyre and the atmosphere discussed here are usually not taken into account. Gyre changes and SH sea ice increase significantly shape and influence the LIG climate in

our H11 simulation. We have shown here that they can only be assessed in a coupled-model simulation framework.

*Code and data availability.*   The source code of the Unified Model (UM) is available under licence. To apply for a licence go to http://www.metoffice.gov.uk/research/modelling-systems/unified-model. JULES is available under licence free of charge, see https://jules-lsm.github.io/. The NEMO model code is available from http://www.nemo-ocean.eu. The model code for CICE can be downloaded from https://code.metoffice.gov.uk/trac/cice/browser.

HadGEM3 model outputs used to support the findings of this study are available from http://gws-access.ceda.ac.uk/public/pmip4/ClimPast_Guarinoetal_2021.

*Author contributions.*   M.V.G ran the HadGEM3 simulations and analysed all simulation results with the contribution of L.C.S., D.S. and J.R. All authors revised the manuscript.

*Competing interests.*   The authors declare that they have no competing interests.

*Acknowledgements.*   Maria-Vittoria Guarino and Louise C. Sime acknowledge the financial support of NERC research grant NE/P013279/1 and NE/P009271/1. The project has received funding from the European Union's Horizon 2020 research and innovation programme under grant agreement No 820970. It is paper number 134. This work used the ARCHER UK National Supercomputing Service (http://www.archer.ac.uk) and the JASMIN analysis platform (https://www.ceda.ac.uk/services/jasmin/).



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
