# Peer review of "The coupled system response to a 250 years of freshwater forcing: Last Interglacial CMIP6-PMIP4 HadGEM3 simulations"

_Climate of the Past, 2021_

## Author Response (AR1)

**Responses to reviewers for "The first 250 years of the Heinrich 11 iceberg discharge: Last Interglacial HadGEM3-GC3.1 simulations for CMIP6-PMIP4 "**

Maria Vittoria Guarino, Louise C. Sime, Rachel Diamond,  Jeff Ridley, David Schroeder

We thank the reviewers for their comments and suggestions. We have edited the manuscript taking into account their recommendations and believe we have addressed their comments in full and to the best of our capabilities.

We also want to express our gratitude to the handling editor and the CP editorial board for granting an extension that accommodated the period of maternity leave of the manuscript's lead author.

Our responses to reviewers can be found below.

**Responses to Reviewer 1**

Major comments

R1: "The authors need to tone down the rhetoric regarding their work being the "the first time that the classical "thermal bipolar seesaw" associated with the H11 event has undiscovered consequences in both Hemispheres". AMOC and the associated climate change have been studied for several decades using water hosing experiments by the climate dynamics community (e.g., Stouffer et al., 2006). The feedbacks from ocean transport and air-sea interactions that are relevant for the AMOC have also been studied extensively (e.g., Buckley and Marshall, 2016; Stocker et al., 2001). Moreover, how the North Atlantic processes can impact the remote regions has also been studied in much detail, including the impact on the tropics and subtropics (e.g., Zhang and Delworth, 2005), global atmospheric teleconnections (e.g., Wu et al., 2013), in a paleoclimate (e.g., Kageyama et al., 2013), and on the Southern Ocean in the context of climate variability in the present-day climate (e.g., Zhang et al., 2017). Given the rich literature on this topic, I think it is unjustified to claim the authors' results are the "first" and show "previously undiscovered consequences in both Hemispheres", unless the authors can show that the AMOC's influences are uniquely tied to the LIG climate and have different dynamics from previous studies. "

We thank the reviewer for the references. Some of them were already used in the manuscript, and we have now included others in Introduction, page 2 lines 36-37 and 55-58, acknowledging the contribution of these relevant studies to the subject. As discussed in the Introduction (lines 41-44, and 55-60), there is a vast literature describing hosing experiments and teleconnections between hemispheres, but less attention has been given to how ocean-atmosphere coupling feeds back into ocean heat transport and sea ice (lines 60-63).

Following the reviewer's recommendations, we have now reworded and toned-down part of the text in Conclusions, page 22 beginning line 402, and abstract, page 1, lines 1-6.

R1: "The sea ice increase over the Southern Ocean in the authors' simulations is interesting and somewhat surprising; more analysis on this is warranted. Is the sea ice increase occurring throughout the year or only in specific seasons? Is the time evolution of the sea ice cover consistent with a SAM-driven mechanism? Can you also show time series of the sea ice expansion? Causal relationship needs to be better established in the authors simulations. Please show the details of the dynamical and thermodynamical processes (Line 242) that drive the sea ice expansion. Are the results model dependent, given the model-dependent responses, e.g., documented in Kageyama et al. (2013)? "

To reply to the reviewer questions we have done the following:

-Figure 12 is now a multi-panel figure showing February (a,b), September (c,d) and Annual (e,f) Antarctic sea ice area. This figure shows that sea ice increases in all seasons, but the increase is more pronounced (and statistically significant) during winter.

-We added new figures, panels b,d and f of Figure 12, where we show timeseries of  H11-LIG sea ice area (SIA) anomaly from year 0 to 250 for annual, February and September sea ice. The time evolution of sea ice anomalies is consistent with a SAM-driven mechanism: SIA anomalies show an overall positive trend over the 250 years of simulation. After about 150 years, anomalies become mostly positive and tend to increase in magnitude. This coincides with the time that 850 hPa winds take to intensity and to move the SAM towards a prevalently SAM-positive regime (as shown in Figure 11).  From Figure 12 we can also see that the positive SIA trend is driven by winter changes (i.e. it is visible in September but not in February), agreeing with what is known about the seasonality of SAM-sea ice interactions. The new figure is discussed in the manuscript at page 15 lines 256-259 and lines 268-278.

-Line 242 referred to our current knowledge of how the SAM can influence sea ice. To bring the reader's attention to the dynamical processes (the ones relevant to the study), we have edited the paragraph starting at line 262 in the revised manuscript describing dynamical processes and providing the relevant references where these processes are better discussed (see page 15, lines 262-267).

-We cannot exclude that our results are to some extent model-dependent. Kageyama et al., 2021 presented a multi-model study for the Northern Hemisphere sea ice using CMIP6 LIG simulations. CMIP6 models simulated an Artic sea ice area of different magnitude (monthly mean SIA differences can be as high as 8 million km2 across models), but all of them were consistent in simulating summer Artic sea ice decrease (compared to PI) during the LIG.

Sea ice formation and melting is affected by large variety of processes in the climate system. It is in fact extremely difficult to unequivocally identify reasons and mechanisms for model spread in sea ice area. Kageyama et al., 2021 also presented a study of some atmospheric and oceanic processes that can impact the representation of sea ice in the HadGEM3, CESM2 and IPSLCM6 models. A similar analysis has not been carried out as today yet for the Southern Hemisphere sea ice. Therefore, while we can expect that the magnitude of the simulated sea ice area will be different across different models, the sign of the anomaly and the exact model behaviour can only be assessed in a multi-model study.

In our hosing simulation, sea ice increases in response to a more positive SAM and thus to atmospheric circulation changes compared to the reference simulation with no freshwater forcing.  As discussed in the paper, the atmospheric circulation changes triggered by the Northern Hemisphere cooling are a rather robust response of the climate system. We can reasonably expect that such behaviour will be replicated in any model that has a fair representation of the atmosphere-ocean coupling. We acknowledge that a multimodel study to confirm and contextualize the results presented in this study would be of much useful and of great interest.

R1: "Where does the deep-water formation take place in the PI and the LIG simulations using HadGEM3? Can the model simulate deep convection over the GIN Seas that has been suggested by observations (e.g., Holte et al., 2017; Lozier et al., 2019)? How are the sites of deep-water formation linked to the heat transport by the AMOC (Figure 9e)? Is it possible that the small heat transport by the AMOC at 50N northward (Line 220–223 & Line 359–361) is caused by the model bias in the deep-water formation over the GIN seas."

Mixed layer depth can be used to detect where deep-water formation takes place within the model. The annual mean mixed layer depth (MLD) for the PI (Menary et al., 2018), LIG and H11 simulation is shown below (Figure 1). Note that we have now added in Supplementary a figure showing H11-LIG MLD anomalies (see response to next point).

[Figure]

Figure 1. Annual mean Mixed Layer Depth for the 500-year long PI simulation (Menary et al., 2018) (top), the LIG (centre) and the H11 (bottom) runs presented in this study.

Present day mixed layer depth and sites of deep convection for the HadGEM3 N96ORCA1 low-resolution model (the version employed in this study) and its higher-resolution version (N216ORCA025) are discussed in details in Kuhlbrodt et al., 2018. In response to the reviewer's questions "Can the model simulate deep convection over the GIN Seas that has

been suggested by observations?", we summarize here the main results found by Kuhlbrodt et al., 2018:

for the N96ORCA1 model, the mixed layer depth tends to be too large in the subtropical latitudes on both hemispheres and is large in the deep-water formation regions of the North Atlantic too.

In particular, in the northern North Atlantic region, N96ORCA1 has a larger than observations mixed layer depth in the central Nordic seas. This behaviour is different from the higher resolution model version N216ORCA025, which shows a deeper mixed layer in larger parts of the Labrador and Irminger Seas.

Thus, the two model versions seem to have different preferred locations for deep-water formation and deep convection is perhaps too strong in the N96ORCA1 model version compared to observations.

Regarding the ocean heat transport in the North Atlantic, Menary et al., 2018 report that for the N96ORCA1 and N216ORCA025 PI simulations ocean heat transport at 26.5N is too weak in both model versions. The AMOC was also found to be too slightly too weak and the upper cell of the AMOC to be too shallow compared to observations.

As for the heat transport at 50N, we are aware of other model biases, such as a weakening of the heat transport due to restricted channels through Denmark and Fram straits, that will have a greater effect in producing weak net heat transport at 50N.

We stress however that while it is not possible to represent the processes mentioned above correctly in a low resolution model (indeed the N216ORCA025 model version has been shown to do better), the analysis presented in our study regards comparison between the LIG and H11 runs in which these biases are equally present.

R1:"A comment related to the above, please consider adding plots of the meridional overturning stream function and the winter maximum mixed-layer depth in the PI, LIG, and H11 simulations. Please also add the mixed-layer depth from observation (Holte et al., 2017) for comparison."

The meridional stream-function anomaly plot has been added to SI (Suppl.Fig.9). We also now show in SI the annual mean mixed layer depth anomaly plot for the Northern Hemisphere (Suppl.Fig.8), this figure was previously mentioned in the paper but not shown (a reference has been added to page 5 line 144). We decided not to add the mixed layer depth from observations, this comparison is not discussed in the paper.

Minor comments

Line 79: change "$N_2O$" to "$N_2O$"
Changed.

Line 109–110: the criteria of p-value should be 0.05 (instead of 0.5), right?
Yes, it was a typo. It has been changed.

Figure 2 and related discussion: how is the North Atlantic and Southern Hemisphere defined spatially? Does the North Atlantic include the Greenland, Iceland, and Norwegian Seas?
Spatial definitions for Northern Hemisphere, Southern Hemisphere and North Atlantic have now been added to Methods section 2.2, see page 4 lines 122-124.

Line 179–182: Please add a mechanistic understanding of the shift of the polar jet in the Southern Hemisphere.
This paragraph was edited and more details were added, see page 9 lines 194-195 of the revised manuscript.

**Figure 5: I do not see "error bars" in the plots.**
Error bars was replaced by "arrows", these are in fact the caps of our error bars which are rather small and difficult to see (hence why the need of using error bar caps).

**Figure 12: is the sea ice concentration annual mean, or for the winter?**
Figure 12 has changed since the previous submission, we now show annual, summer and winter sea ice maps.

**Responses to Reviewer 2**

R2: "In general, 350 y, which is the time that the LIG simulation was run, is not enough time to equilibrate the physics of the deep ocean. See for example Marzocchi and Jansen (2017, GRL), where they have to run CCSM4 with LGM boundary conditions for several thousand years to achieve physical equilibrium in the deep ocean. The authors should give evidence that their LIG simulation is in equilibrium, or explain why such requirement is not needed in this study."

We fully agree with the reviewer that 350 model years cannot be enough for the deep ocean to reach equilibrium. We stress however that coupled climate models, like the ones used for CMIP simulations, are unlikely to ever be able to reach full oceanic equilibrium. As for HadGEM3, 8000+ years is the latest estimate we are aware of (through informal communications) for the time that one should run the NEMO model only to spin-up the deep ocean.

The study we present here comes after previous publications form our research group and the wider UK CMIP6 and PMIP4 communities, all referenced in the manuscript, where simulations protocols and technical aspects of the model set-up are discussed in greater details. In particular, the LIG simulation was spun-up for 350 model years, at which point the model had reached a quasi- atmospheric and upper-ocean equilibrium. As pointed out in the manuscript (see section 2.1, beginning line 85), Williams et al., 2020 presented a detailed analysis of what criteria were used to assess the system equilibrium. Further to this, in Guarino et al., 2020 an analysis concerning the HadGEM3 model internal variability and what simulation length can be sufficient for studying climate features can be found.

We have reworded somewhat lines 93-95, but given the published literature on the topic, cited in the manuscript, we do not feel that adding further details on the technical aspects of the PMIP4 simulations would be beneficial to the study.

R2: "Concerning the 250 y H11 simulation, it is necessary to distinguish between the 250 y of a computer simulation and the first 250 y of H11 in climate history. If the authors want to claim that their 250 y simulation represents the actual first 250 y of H11 they should give evidence of the correct representation of the timing of physical processes in the atmosphere and ocean. If this can't be done, then I would switch the title to "A 250 yea simulation of the Heinrich 11 iceberg discharge: Last Interglacial HadGEM3-GC3.1 simulations for CMIP6-PMIP4", and all suggestions in the manuscript that this is an actual simulation of the first 250 y of H11 should be changed accordingly."

We accept the reviewer's suggestion and have changed the manuscript title accordingly. It is indeed correct that we present a 250 years of hosing experiment under LIG climate conditions.

R2: "In several places of the manuscript the authors mention that a longer simulation cannot be performed due to computational constrains. However, conjectures are made

about the evolution of some processes beyond the length of the H11 experiment (e.g., lines 255 and 295). Could a 500 y H11 experiment be performed before the publication of this paper? I think it would benefit the science shown, increase the relevancy of this work, and would be beneficial for the science community. In its current state, the analysis is limited to a transient 250 y run, and not all processes governing the H11 climate have manifested (according to the authors). In addition, little evidence is shown that the 250 y presented in the manuscript is a correct representation of what happened during the onset of H11, and not just an abstract modelling study (see my comment about correct representation of timing). "

Since the first submission, the H11 simulation used in this study has been extended by further 100 years (we remind here that HadGEM3 is not a fast-running climate model). The manuscript 's lead author is not the person in charge of the new run, but results for the extended run have been kindly made available to us and are presented here to address the reviewer's comments. Below we show the time evolution of the maximum meridional stream-function at 26N (Figure 2), the ocean potential temperature (Figure 3), the SAM index (Figure 4), and Antarctic sea ice (Figure 5 and 6) from year 0 to 350 of simulation. After further 100 years of simulation, the model behaviour does not change, and the main conclusions drawn and discussed in the manuscript remain valid. These are:

-after about 150 years the AMOC reaches a plateau (Figure 2);

-Southern Ocean keeps on warming (Figure 3);

-the SAM index continues its upward trend and becomes more and more positive as time goes by (Figure 4);

-Antarctic sea ice continues to increase (Figure 5 and 6).

As the reviewer correctly says, we speculate on a number of possible scenarios that could come about if the simulation was run for longer. We clearly state that these are speculations (see page 21, 23 lines 355,405), because the timescales over which different climate responses might manifest are much larger than a few hundred (additional) years. We explain this at page 21, lines 348-352, where we mention changes in sea ice due to deep ocean circulation changes, and page 21 lines 344-347 where we mention possible atmospheric responses to a different pattern of ocean warming once the Southern Ocean has warmed (we estimate in the study the need of about 1500 years for the SO to reach the 2-3 degree warming found in climate proxy records).

To (partially) address the reviewer's comment we have now added a paragraph to Conclusions, see page 23 lines 409-415, acknowledging that the H11 simulation is currently being extended and that results from the extended run will be object of further studies. We point out that by looking at the additional 100 years available no different conclusions could be drawn, and that the different climate responses mentioned in the study are likely to manifest on much longer timescales because of their dependency on the ocean response.

[Figure]

Figure 2. Maximum meridional stream-function at 26N for LIG (orange) and H11 (blue). Thin lines are annual means, thick lines are 11-year running means.

[Figure]

Figure 3. Hovmoller diagram of SH sea water potential temperature H11-LIG anomalies at 500m depth.

[Figure]

Figure 4. The annual SAM index for the H11 simulation. The linear regression line (dashed) represents a statistically significant positive trend with pvalue < 0.05.

[Figure]

Figure 5. Southern Hemisphere sea ice area time-series for the H11 simulation. Thin lines are monthly means, thick lines are 10-year means.

[Figure]

Figure 6. Maps of H11-LIG sea ice concentration anomalies for: last 100 years of H11 run (left), first 50 years of extended run (centre) and last 50 years of extended run (right).

R2: "Lines 15-20: The authors should better describe the LIG and H11 climate periods. What was the length of LIG, and of H11? Did H11 occur at the onset of the LIG, or somewhere in the middle? This is important in order to put the reader in context of the climate period we are discussing."

We now specify the temporal intervals for the LIG period and the H11 event, page 1 lines 17 and 19.

R2: "Lines 16-24: "warming of the Southern Ocean during this time is attributed to a slowdown of the Atlantic Meridional Overturning Circulation (AMOC), which has been suggested as a mechanism to explain the 2-3°C Southern Ocean warming found in Southern Ocean and Antarctic climate records" Are the papers mentioned to justify this statement modeling- based or data-based? A brief description of them would be important to understand the scientific basis of this work."

These cited studies all report estimates for Southern Ocean and Antarctic surface air temperatures based on proxy records of various nature (e.g., ice cores, marine cores). We have added further details regarding the cited papers at pages 1,2 lines 23-28.

R2:"Line 79: N2O not N20."

This has been changed.

R2: "Line 125: The relation between heat transport and AMOC in interesting. However, a more complete analysis should include salt fluxes as well. How do they change between the LIG and H11 simulations? Do they have any relevancy for AMOC? How are they affected by the North Atlantic hosing?"

Northern Hemisphere (NH), Southern Hemisphere (SH) and Global salinity for the H11 and LIG simulations have been added to Supplementary Information for completeness and are referenced in the paper at page 5, line 144. Despite some work in literature (e.g. Cael et al., 2020) suggesting that, for present-day conditions, an increase in freshwater fluxes might actually increase the AMOC strength by enhancing the salinity gradient in the surface and deep north Atlantic water, in our simulation (and in other studies using HadGEM3, see e.g. Jackson and Wood, 2018) the net impact of freshwater fluxes is to decrease the AMOC (see Figure 1).

The salinity of the global ocean is much lower in H11 than LIG (Suppl.Fig.10), this change is driven by the freshening of the North Atlantic (NA). Suppl.Fig.11 shows NA and SH salinity for the top 200m and the top one kilometre of the water column (similarly to Figure 2 in the manuscript). In the North Atlantic, H11 salinity is much lower than the LIG salinity everywhere. In the SH, salinity increases slightly- changes are one order of magnitude smaller than NH changes.

Higher southern hemisphere sea ice concentration and a warmer Southern Ocean can both play a role in increasing ocean salinity in the Southern Hemisphere. Thus the salinity changes in Suppl.Fig.11 might be consistent with the analysis proposed in the manuscript. However, a thorough analysis of salinity changes would require looking at precipitation changes too. This is beyond the scope of our study, and we therefore decided to not include salinity results in the manuscript and present them in Supplementary Information only.

R2:"Line 195: Could you show the changes in meridional heat transport for the other ocean basins? (in the supplement is OK) Although this work focuses on the Atlantic, effects in other ocean basins may be relevant for readers."

The advective heat transport for the Pacific and Indian oceans is now shown in SI (Suppl.Fig.12) and referenced in the manuscript a page 14, line 244.

R2: "Line 208: In Figure 10 I don't see any subtropical gyre intensification for H11."

The intensification of the H11 subtropical gyre is visible in its southern branch, western quadrant. This is carefully described at page 14, lines 225-228:

"The southern branch of the subtropical gyre is also affected by wind changes. The barotropic streamfunction for the H11 subtropical gyre strengthens in the 20-30N region (particularly in the western quadrant) because of the modest acceleration in the easterly trade winds."

R2: "Lines 249-255: Could the absence of a dipole in the Antarctic sea ice response to a positive SAM be due to issues with the sea ice model, like for example lack of resolution or limitations in the parametrization of processes? Please expand in the paper. Could you show the sea ice in winter months in the supplementary material?"

Sea ice discussion has been expanded in the revised manuscript, also in response to reviewer 1. Please see the new discussion at page 15, lines 257-278 and the modified Figure 12.

Regarding possible limitations in modelling sea ice due to the need of parameterizing sea ice physics, HadGEM3 uses the most advanced sea ice model currently available. While it is impossible to exclude that model uncertainties can impact our results, we have no indication that the sea ice behaviour discussed in our study is caused by sea ice model shortcomings (see Ridley et al. 2018, where the sea ice model is presented).

REFERENCES

Kageyama et al., 2021: A multi-model CMIP6-PMIP4 study of Arctic sea ice at 127 ka: sea ice data compilation and model difference. Climate of the Past, 17,37-62.

Kuhlbrodt et al., 2018.: The low resolution version of HadGEM3 GC3.1: Development and evaluation for global climate. J. Adv. Model. Earth Sy., 10, 2865–2888.

Menary, Matthew B., et al. "Preindustrial control simulations with HadGEM3-GC3. 1 for CMIP6." Journal of Advances in Modeling Earth Systems 10.12 (2018): 3049-3075.

Cael, B. B., and Malte F. Jansen. "On freshwater fluxes and the Atlantic meridional overturning circulation." Limnology and Oceanography Letters 5.2 (2020): 185-192.

Jackson, L. C. and Wood, R. A.: Timescales of AMOC decline in response to fresh water forcing, Climate Dynamics, 51, 1333–1350, 2018b.

Ridley, J. K., Blockley, E. W., Keen, A. B., Rae, J. G., West, A. E., and Schroeder, D.: The sea ice model component of HadGEM3-GC3. 1, Geoscientific Model Development, 11, 713–723, 2018.